# XGBoost-Based Simple Three-Item Model Accurately Predicts Outcomes of Acute Ischemic Stroke

**DOI:** 10.3390/diagnostics13050842

**Published:** 2023-02-22

**Authors:** Chen-Chih Chung, Emily Chia-Yu Su, Jia-Hung Chen, Yi-Tui Chen, Chao-Yang Kuo

**Affiliations:** 1Department of Neurology, Taipei Medical University—Shuang Ho Hospital, New Taipei City 235, Taiwan; 2Department of Neurology, School of Medicine, College of Medicine, Taipei Medical University, Taipei City 110, Taiwan; 3Taipei Neuroscience Institute, Taipei Medical University—Shuang Ho Hospital, New Taipei City 235, Taiwan; 4Graduate Institute of Biomedical Informatics, College of Medical Science and Technology, Taipei Medical University, Taipei City 110, Taiwan; 5Clinical Big Data Research Center, Taipei Medical University Hospital, Taipei City 110, Taiwan; 6Smart Healthcare Interdisciplinary College, National Taipei University of Nursing and Health Sciences, Taipei City 112, Taiwan; 7Department of Health Care Management, College of Health Technology, National Taipei University of Nursing and Health Sciences, Taipei City 112, Taiwan; 8Department of Education and Research, Taipei City Hospital, Taipei City 103, Taiwan

**Keywords:** glucose, machine learning, NIHSS, prognosis, acute ischemic stroke, XGBoost

## Abstract

An all-inclusive and accurate prediction of outcomes for patients with acute ischemic stroke (AIS) is crucial for clinical decision-making. This study developed extreme gradient boosting (XGBoost)-based models using three simple factors—age, fasting glucose, and National Institutes of Health Stroke Scale (NIHSS) scores—to predict the three-month functional outcomes after AIS. We retrieved the medical records of 1848 patients diagnosed with AIS and managed at a single medical center between 2016 and 2020. We developed and validated the predictions and ranked the importance of each variable. The XGBoost model achieved notable performance, with an area under the curve of 0.8595. As predicted by the model, the patients with initial NIHSS score > 5, aged over 64 years, and fasting blood glucose > 86 mg/dL were associated with unfavorable prognoses. For patients receiving endovascular therapy, fasting glucose was the most important predictor. The NIHSS score at admission was the most significant predictor for those who received other treatments. Our proposed XGBoost model showed a reliable predictive power of AIS outcomes using readily available and simple predictors and also demonstrated the validity of the model for application in patients receiving different AIS treatments, providing clinical evidence for future optimization of AIS treatment strategies.

## 1. Introduction

Stroke is the second leading cause of death and the third leading cause of disability worldwide [1,2]. The global burden of stroke has increased substantially over the past 30 years owing to population growth, aging, and exposure to risk factors such as high plasma levels of fasting glucose. Among all stroke types, ischemic strokes account for the largest proportion of all new strokes. In 2019, there were an estimated 7.6 million new ischemic strokes globally, and the economic burden of stroke is estimated to be more than US$ 721 billion [1]. In addition to the high incidence of stroke occurrence, recurrence, and death, ischemic stroke may cause various functional impairments, cognitive impairment, or post-stroke depression, contributing to the high prevalence of disability in stroke survivors [1,2,3,4,5]. In the report by Yao et al. [4], two-thirds of patients with ischemic stroke had sequelae symptoms, and nearly one-fourth had varying degrees of disability. Consequently, stroke has a wide range of negative physical and economic impacts on patients and their families. Metabolic factors such as high blood pressure, high blood sugar, and high cholesterol are strongly linked to ischemic stroke. These potentially modifiable risks are associated with outcomes after the episode of ischemic stroke [1,2,3].

Various scoring systems have been developed to predict the prognosis and treatment outcomes after acute ischemic stroke (AIS) [6]. However, the clinical application of these models is often limited due to the complexity of the scoring methods and their mild-to-moderate prediction accuracy [6].

Machine learning methods provide a means to process large amounts of data and enhance data interpretation [7,8,9,10]. In recent years, exponential advances in computing power and improvements in algorithms have led to the successful application of machine learning algorithms in developing clinical prediction and decision systems for AIS [9,10,11,12,13,14,15,16]. Gradient tree boosting is a widely-used technique among machine-learning algorithms which has been proven to provide state-of-the-art performance in classification in clinical practice [17].

Extreme gradient boosting (XGBoost) is a tree-boosting ensemble algorithm that gradually establishes the losses produced by weak learners based on a decision tree [18,19]. Having a higher predictive performance than other algorithms, the XGBoost algorithm has provided model interpretation in many fields, including medical and environmental research [18,19,20,21]. Previously, Patel et al. used XGBoost to predict the mortality of patients in the intensive care unit using a multi-center database [18]. Additionally, Zhang et al. indicated that the predictive performance of XGBoost was better than that of logistic regression (LR) in distinguishing between patients with and without a response to fluid intake in urine output [20].

In stroke-related studies, XGBoost performed better at predicting 30-day readmissions than other machine learning algorithms [22]. When used to predict 90-day readmissions for patients with AIS, XGBoost outperformed the LR model [23]. In assessing patient outcomes following stroke, XGBoost showed the best performance among all classification models, achieving an accuracy of 82% based on ten activities of daily living tasks, measured using wearable technologies [24].

Although machine learning approaches have expanded the field of diagnostic and predictive tool design, many models require embedding of overly complex variables, making data collection and organization complicated, thus potentially limiting their clinical applicability [9,10,13,25]. A previous study applied machine learning models for ischemic stroke using data from 35,798 patients with 17 predictive features [13]; in addition, the machine learning-based models developed by Heo et al. used 38 variables to predict AIS outcomes [10]. Both studies also included different predictors, such as clinical variables and brain images, but without a universal modality [9,10,13,16,25].

Although challenging, an all-inclusive predictive model that utilizes simple, easily accessible predictors for patients with AIS with different clinical characteristics and receiving different treatments will be crucial for clinical decision-making and treatment strategies. In previously proposed prediction models, factors, such as age, fasting glucose, and stroke severity (as represented by NIHSS scores), served as general, and often overlapping, independent predictors of AIS functional outcomes [6,26,27]. The current study hypothesized that age, fasting glucose, and NIHSS score could serve as simple, generic, and readily available factors and provide sufficient predictive power to predict outcomes after stroke. Thus, the study aimed to develop predictive models using these most general predictors, utilizing the advantages of machine learning algorithms in analyzing large amounts of nonlinear data.

The present study (i) provided accurate outcome predictions for AIS by applying XGBoost-based models using three simple parameters: age, fasting glucose, and NIHSS score, and (ii) explored the clinical validity of the current model in patients with AIS who received different treatments, including intravenous thrombolysis (IVT), endovascular therapy (EVT), and non-thrombolytic treatments.

## 2. Materials and Methods

### 2.1. Participants

In the present study, we retrospectively reviewed the medical records of patients diagnosed and managed at Shuang-Ho Hospital between January 2016 and March 2020. Patient records were retrieved from the Taiwan Stroke Registry, a multi-center database that collected clinical data from patients with AIS across major hospitals and health facilities in Taiwan [28,29].

The inclusion criteria for this study were patients aged ≥18 years who were admitted to the hospital for the diagnosis and treatment of AIS. AIS was defined as the acute onset of neurological deficits with signs or symptoms and presenting to the hospital within 10 days of onset. In all patients, either a non-contrast computed tomography (CT) scan of the head or magnetic resonance imaging (MRI) of the brain was performed at the time of admission. The exclusion criteria were as follows: patients presenting with acute intracranial hemorrhage or subarachnoid hemorrhage on admission, lack of fasting glucose or NIHSS records on admission, no modified Rankin Scale (mRS) score recorded at three months, and patients transferred to other hospitals. These were excluded to ensure availability and consistency of records.

Demographic data were collected at admission, including age, sex, NIHSS score, and treatment for AIS (i.e., IVT, EVT, and non-thrombolysis). IVT and EVT treatments followed the American Heart Association/American Stroke Association guidelines [30]. The patients’ vascular risks and comorbidities, including hypertension, diabetes mellitus, hyperlipidemia, atrial fibrillation, previous stroke or transient ischemic attack (TIA), and ischemic heart disease, were also documented.

Fasting glucose levels were measured within 72 h of admission. The mRS scores were used to assess functional outcomes after AIS at three months. NIHSS scores at admission and mRS scores at 3 months were assessed by a certified stroke specialist. For patients treated with IVT and EVT, head CT/MRI was repeated within 24 h to rule out treatment-related intracranial hemorrhage [30]. Two independent neurologists and a radiologist interpreted all the CT/MRI findings. A favorable outcome was defined as an mRS score of 0 or 1, whereas an unfavorable outcome was defined as an mRS score ≥2 at the 3-month follow-up [31,32].

### 2.2. Statistical Analyses

All statistical analyses were performed using the SAS 9.4 software (SAS Institute Inc., Cary, NC, USA). The variables were summarized using descriptive statistics. All continuous variables were expressed as mean and standard deviation. Fisher’s exact test was used to determine associations between the two categorical variables in patients with favorable and unfavorable outcomes. Student’s *t*-test was used to determine the statistical significance of differences between the means of two continuous variables. We used receiver operating characteristic (ROC) curves to assess model performance and calculated each predictive model’s accuracy, sensitivity, and specificity, as well as the area under the ROC curve (AUC). Statistical significance was set at a two-tailed *p*-value of < 0.05.

### 2.3. Application of Extreme Gradient Boosting (XGBoost) Modeling

All XGBoost models were developed using the “caret” and “xgboost” packages in RStudio software (version 1.3.1073; 2009–2020 RStudio, PBC). XGBoost algorithm was used to train and predict 3-month unfavorable outcome.
(1)$$Predicted\;outcome={\hat{y}}_{i}=\overset{K}{\underset{k=1}{\sum }}f_{k}\left(X_{i}\right),\;f_{k}\in F$$

In the process of training prediction model, K trees were generated. In XGBoost, the set of functions used in the model was minimized by the following regularized objective:(2)$$L(\phi )=\sum_{i}l(({\hat{y}}_{i}),\;y_{i})+\sum_{k}\Omega (f_{k})$$
where $\Omega (f_{k})=\gamma T\;+\;\frac{1}{2}\;\lambda {\left||\omega |\right|}^{2}$, *l* is a differentiable convex loss function measuring the predicted outcome ${\hat{y}}_{i}$, and the real outcome *y_i_*. Considering that the tree with more depth smoothed the final learned weights, the prediction model was penalized by the regularization term Ω, used to avoid overfitting.

XGBoost did not generate all trees at one time. It began from the 0-th tree. It split each node in the tree; the original node continued splitting to right and left nodes. After a new node splits, it was essential to check whether the new split would give Gain to the loss function. The variable and its point with maximum value of Gain were selected.

The final information, Gain, of the objective function after each split was:(3)$$\mathrm{Gain}=Obj_{L+R}-\left(Obj_{L}+Obj_{R}\right)=\frac{1}{2}\left[\frac{{\left(\;\sum_{i\;\in \;I_{L}\;}^{}\mathrm{gi}\right)}^{2}}{\;\sum_{i\;\in \;I_{L}\;}^{}\mathrm{hi}+\lambda }+\frac{{\left(\;\sum_{i\;\in \;I_{R}\;}^{}\mathrm{gi}\right)}^{2}}{\;\sum_{i\;\in \;I_{R}\;}^{}\mathrm{hi}+\lambda }-\frac{{\left(\;\sum_{i\;\in \;I\;}^{}\mathrm{gi}\right)}^{2}}{\;\sum_{i\;\in \;I\;}^{}\mathrm{hi}+\lambda }\right]-\gamma$$
where, *L* and *R* are the subsets of left and right nodes after splitting from the instance set *I*. γ is a splitting threshold for suppressing the growth of the tree [17,33]. It was an important metric, used to rank the importance of variables in this study.

### 2.4. Ten-Fold Cross-Validation

Ten-fold cross-validation was applied in this study. Library caret was used for cross-validation and to generate the measures. The dataset was randomly divided into ten subsets, with nine datasets used as training data to develop the prediction model, and the remaining subset used to validate the model. Furthermore, optimal tuning hyperparameters were selected using ten-fold cross-validation of the training data. The hyperparameters assigned for optimization in the final prediction model were as follows: “nrounds,” “max_depth,” “eta,” “gamma,” “colsample_bytree,” “min_child_weight,” and “subsample.”

### 2.5. Ranking the Importance of Variables

The metric Gain was used to evaluate the importance of each variable generated by the binary prediction model based on XGBoost. Gain is the measure that enhances accuracy by a specific variable to the branch, as classified errors can be decreased by adding a particular variable, and the two new branches become more accurate. A higher percentage indicated that the variable was more important than the others [34].

### 2.6. Partial Dependence Plots (PDPs)

Visualization is a reasonable interpretation method for black-box machine learning algorithms, illustrating the relationship between the outcomes and variables selected in the prediction model [35]. PDPs represent one such method and were used to depict the change in the outcome of interest in the prediction model (the marginal effect) while the specific variable changed [36]. Package “pdp” generated PDPs for each variable in the R environment.

## 3. Results

We used the packages in the R environment. Ten-fold cross-validation was the validated method in the package “caret”, which could then be used to find the optimal hyperparameters for the prediction model in terms of AUC. The optimal hyperparameters were decided during the ten-fold cross-validation and incorporated while building the prediction model based on the XGBoost algorithm. We used the Gain indicator to rank the importance of the variables in different treatments. To find the relationship between 3-month unfavorable outcomes and each variable, partial dependence plots were used to depict the propensity in all variables.

### 3.1. The Structure of Study

The analytical flowchart of this study is shown in Figure 1. Data from 2384 participants were collected. After the completion of screening data, the records of 1848 patients were analyzed. There were several hyperparameters that needed be controlled. To achieve the best predictive performance, hyperparameters had to be optimal. Ten-fold cross-validation was used to find the optimal hyperparameters and improve the prediction model. Finally, the ranking importance of variables and PDPs were generated by the final prediction model to uncover the relationship between 3-month unfavorable outcomes and variables.

### 3.2. Demographical and Baseline Characteristics of the AIS Cohort

The cohort included 706 females and 1142 males, with a mean age of 68.29 (±13.66) years and a mean baseline NIHSS score on admission of 7.58 (±7.85) (Table 1). On admission, 213 patients (11.53%) underwent IVT, with or without EVT. Among these, 180 patients (84.51%) who received IVT only were termed the IVT group. Among the entire cohort, 100 patients (5.41%) who received EVT were designated as the EVT group, including 33 patients who received EVT following IVT and 67 patients who received EVT only. Three months after AIS, 52.38% of the patients had an unfavorable outcome. Patients with unfavorable outcomes were significantly older, had higher NIHSS scores, and had higher fasting glucose levels than those with favorable outcomes. Furthermore, in vascular risk factors, Fisher’s exact test also showed that diabetes mellitus, hyperlipidemia, atrial fibrillation, previous stroke or TIA, and ischemic heart disease were associated with 3-month outcomes, as was the gender of patients (Table 1).

### 3.3. Model Performance

In the current study, we incorporated fasting glucose levels, age, and NIHSS as inputs to predict patients’ potential unfavorable outcomes three months post-stroke using the XGBoost algorithm.

The mean and standard deviation for accuracy, sensitivity, specificity, and AUC of the predictive model, based on XGBoost with ten-fold cross-validation, are shown in Appendix A. The accuracy, sensitivity, and specificity were 0.7654, 0.7886, and 0.7644, respectively, with an AUC of 0.8595. Figure 2 shows the ROC curve of the validation sets during ten-fold cross-validation. The AUC values of the model, applied in patients who received non-thrombolytic, IVT, and EVT treatments, were 0.8615, 0.8423, and 0.7733, respectively.

The optimal tuning hyperparameters were also provided via ten-fold cross-validation. The hyperparameters were selected to train the final model, as follows: “Nround = 50”, “max_depth = 1”, “eta = 0.3”, “gamma = 0”, “colsample_bytree = 0.8”, “min_child_weight = 1”, and “subsample = 0.5”.

Figure 3 illustrates the importance of predictors for the entire cohort and patients receiving different treatments. The importance was measured by Gain, which was generated from the package “xgboost” from the ranking based on Gain. In both the whole cohort and in patients who received IVT or non-thrombolytic treatment, the NIHSS score was the most crucial variable for predicting AIS outcome. In the EVT group, the most important predictor was fasting glucose level, with the NIHSS score being less important than age.

Figure 4 depicts the partial dependence plot of the models, showing the relationship between unfavorable outcomes and each variable. Generally, the three predictors in this model offered a positive nonlinear effect on unfavorable outcomes in AIS. The NIHSS score significantly impacted outcomes: the higher the NIHSS score, the worse the prognosis was at three months after the stroke. As predicted by the XGBoost model, when the initial NIHSS score was >5, patients were likely to have an unfavorable prognosis after AIS (Figure 4a). In addition, as shown in Figure 4b, patients over 64 years of age were more likely to have adverse outcomes after AIS in our model, and fasting blood glucose > 86 mg/dL was also associated with a high risk of adverse outcomes (Figure 4c).

### 3.4. Comparison of the Patients with and without IVT in the EVT Group

To further interpret the current model for the EVT group, we compared the characteristics of patients in the EVT group with and without IVT and assessed the importance of predictors of the two subgroups separately. When comparing the EVT, with and without IVT subgroups, there were no differences between age (67.12 (±12.48) vs. 70.28 (±13.75), *p* = 0.2535), fasting glucose (184.15 (±56.61) vs. 168.60 (±58.16), *p* = 0.2049), or NIHSS scores (18.15 (±6.60) vs. 19.04 (±7.60), *p* = 0.5473). There were also no differences in the outcomes by mRS assessment between the two subgroups three months after the stroke (*p* = 1.000).

Figure 5 demonstrates the importance of predictors for the EVT group with and without IVT treatment. Similar to the non-thrombolytic and IVT groups, the NIHSS score was the most important predictor of prognostic relevance in the EVT with IVT subgroup (Figure 5a). However, in the subgroup of EVT without IVT, fasting glucose was determined as the most crucial factor in predicting outcomes in the XGBoost model (Figure 5b).

## 4. Discussion

In this study, we applied an XGBoost-based model to predict the neurological prognosis of patients with AIS three months after stroke based on three simple predictors: age, NIHSS score on admission, and fasting glucose levels. The generated XGBoost model achieved a good validation performance, with an AUC of 0.8595 and validation accuracy, sensitivity, and specificity values of 0.7862, 0.7886, and 0.7644, respectively. Using the model developed by the XGBoost algorithm, we evaluated the importance of each predicting feature by Gain. In our model, NIHSS was the most critical variable for predicting unfavorable outcomes after AIS, with initial NIHSS scores above 5 eliciting a higher propensity for unfavorable outcomes. For patients treated with non-thrombolytic therapy, IVT, and EVT, our models achieved AUCs of 0.8615, 0.8423, and 0.7733, respectively. Compared to previous studies, our model simplified the variables needed to predict the prognosis of AIS and could be applied to patients receiving different AIS treatments.

Globally, stroke remains the leading cause of death and disability, and the disease and medical burden of stroke have continued to increase over the past three decades [1,2,3]. Therefore, the development of reliable prognostic prediction methods for AIS is necessary to improve disease treatment decisions, allocation of care resources, and predictive treatment strategies.

Our results suggested that aging had a negative impact on long-term treatment outcomes following AIS. Evidence suggested that older patients exhibited more severe AIS symptoms than younger patients and were associated with more unfavorable clinical outcomes, such as higher rates of disability and mortality [37,38]. The strong association between aging and unfavorable AIS outcomes, independent of stroke severity, characteristics, or comorbidities, supported the widespread use of age as a factor in various scoring systems, and its characterization as a highly-weighted covariate [6,26].

Stroke severity is another widely-used variable in AIS outcome prediction scoring systems [6,26]. The NIHSS is a 15-item neurological examination scale with higher scores indicating higher stroke severity [39]. Owing to its simplicity, reproducibility, reliability, and validity of estimation scales extracted from medical records, the NIHSS score is a commonly used predictor of AIS outcomes in stroke predictive models [6,26]. Previous studies have reported that the NIHSS score on admission was a significant factor in the development of unfavorable outcomes based on multivariate logistic regression models [40,41,42]. Consistent with Al Khathaami et al. [40], our data showed that patients with NIHSS scores > 5 on admission were more likely to develop an unfavorable prognosis. In addition, our model demonstrated more detailed probabilities using partial dependency plots of NIHSS scores and rationalized the use of the NIHSS score as a crucial predictor in the XGBoost-based prediction model.

Hyperglycemia after stroke may adversely affect clinical outcomes [30]. Previous studies confirmed that hyperglycemia on hospital admission in patients with AIS was associated with greater final infarct size, worse clinical prognosis, and higher mortality rates [15,16,43,44]. Furthermore, a retrospective study conducted by Sung et al. [45] demonstrated that, compared with random glucose or HbA1c levels, fasting glucose was a stronger independent predictor of unfavorable neurological outcomes in patients with AIS. Unlike age and NIHSS scores, which are determined at the time of AIS presentation [46], blood glucose is a modifiable factor. With proper glycemic control, it is possible to improve the prognosis of patients and reduce the negative physical impact of AIS [30]. These findings, coupled with the general applicability of glucose levels in various AIS outcome prediction models, rationalized the decision to use fasting glucose as a predictor in our prediction model [15,16,26,47,48].

Furthermore, in the present XGboost model, fasting glucose was assessed as the most critical predictor of outcomes for patients who received EVT. The result was in concordance with current evidence showing that impaired fasting glucose after AIS onset to be an independent risk factor for 3-month unfavorable outcomes after EVT, especially for patients aged ≥60 years [49,50]. The knowledge gained from using machine learning on clinical data also implied that applying these insights could impact future clinical practice, such as the optimal range of glycemic control for patients receiving different AIS treatments.

The complexity of the scoring methods and the low predictive accuracy of the models make it difficult for many scoring systems or models to be used in clinical practice to predict AIS prognosis. In this context, our model used readily available patient information, such as age, NIHSS score, and fasting glucose, providing good predictive accuracy. This gives our model the potential to be suitable to assist in clinical applications. For patients with AIS, age and stroke severity were determined at the time of stroke onset. However, clinically modifiable factors, such as blood glucose, in predictive models, could aid clinical decision-making and improve patient prognosis. For instance, one could try to optimize a patient’s glycemic control and apply the model to evaluate whether it would provide the patient with a better outcome after AIS.

Stroke prognosis is influenced by various clinical factors and treatment approaches [30,32,51]. Moreover, the multifactorial and complex nature of AIS limits the predictive performance and reliability of traditional predictive models and scoring systems to individualize treatment. Our study showed that the application of XGBoost was able to accurately predict AIS outcomes using simple inputs. The predictive power of our model was consistent with the documented application of machine learning algorithms to assist in the diagnosis of stroke and predict its outcome [16,52,53]. For patients with a predicted high risk of unfavorable prognosis, our model could facilitate the timely adoption of personalized treatments, highlighting its potential to help stratify patients, individualize treatment, and optimize the therapeutic management of patients with AIS.

The present study had several limitations. First, the data analyzed in this study were limited to patients with AIS from a single stroke center. Therefore, a larger cohort from multiple centers will be required to validate and generalize our model. Second, the model was based on only three clinical predictors and did not include data obtained from neuroimaging or other neurophysiological examinations [44], thus potentially limiting its predictive accuracy. Third, as the dataset was extracted from clinical registry data, there was a relatively modest proportion of missing data for baseline variables, which may have also affected the accuracy of the model. Finally, as the treatments for AIS continue to advance, the treatment techniques (e.g., EVT) and their respective clinical guidelines continue to be updated. As time progresses, new treatments and technologies could significantly improve the prognosis of AIS patients. Therefore, predictive models based on past patient information may have limited abilities for prospective patients with AIS.

## 5. Conclusions

In the present study, we developed an accurate and reliable model to predict outcomes at three months following AIS by applying XGBoost-based machine learning algorithms, using three predictors: age, fasting glucose, and NIHSS score. The generated XGBoost model achieved a notable validation performance with an AUC of 0.8595. This model also demonstrated its validity in patients with AIS who received different treatments, including IVT, EVT, and non-thrombolytic therapy. In the whole cohort and patients who received IVT or non-thrombolytic treatment, the NIHSS score was the most crucial predictor of AIS outcome. In the EVT group, the most important predictor was fasting glucose. Our results showed the potential clinical applicability of this model to facilitate individualized risk prediction and aid the formulation of diagnostic decisions for patients with AIS. Based on these results, future work is being shaped to develop predictive models that incorporate a more diverse patient population and can be automatically updated with new patient information and future advancements.

## Figures and Tables

**Figure 1 diagnostics-13-00842-f001:**
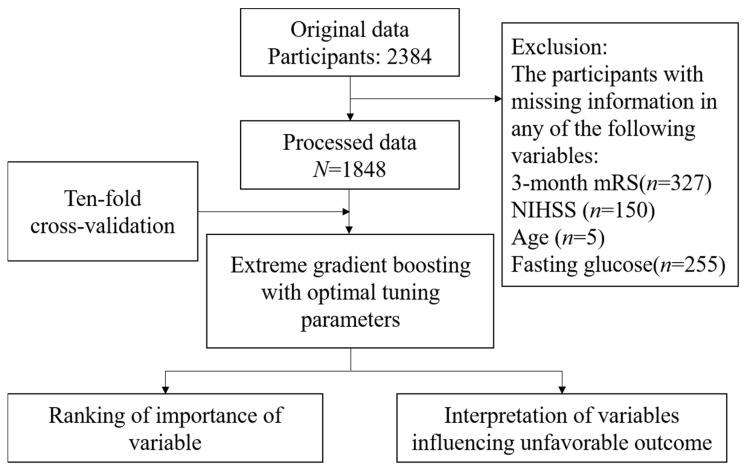
Analytical flowchart of development of the model for predicting outcome three months after stroke.

**Figure 2 diagnostics-13-00842-f002:**
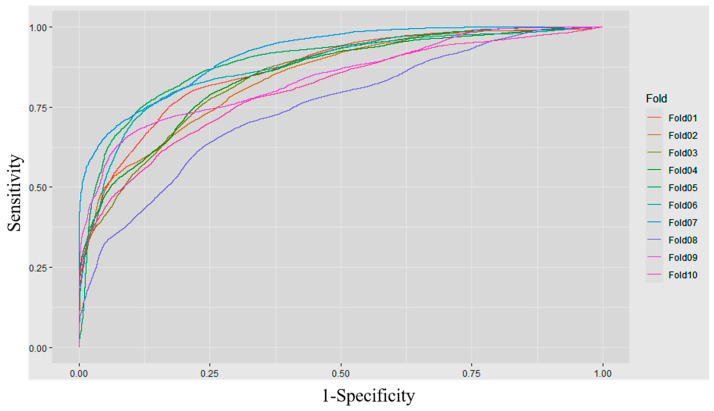
ROC curves of ten-fold cross-validation based on XGBoost.

**Figure 3 diagnostics-13-00842-f003:**
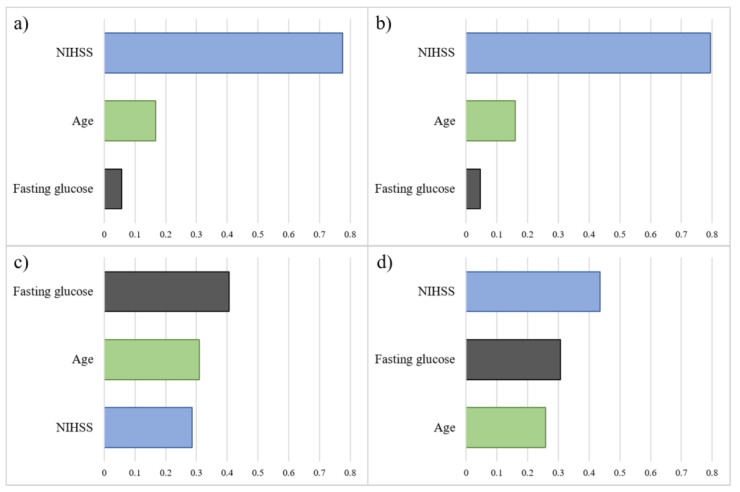
The importance and the ranking of the predictors based on Gain in XGBoost. Following the increasing accuracy of the branches, the longer horizontal bar represents greater importance attributed to the model. (**a**) Entire cohort. (**b**) Non-thrombolytic. (**c**) EVT. (**d**) IVT.

**Figure 4 diagnostics-13-00842-f004:**
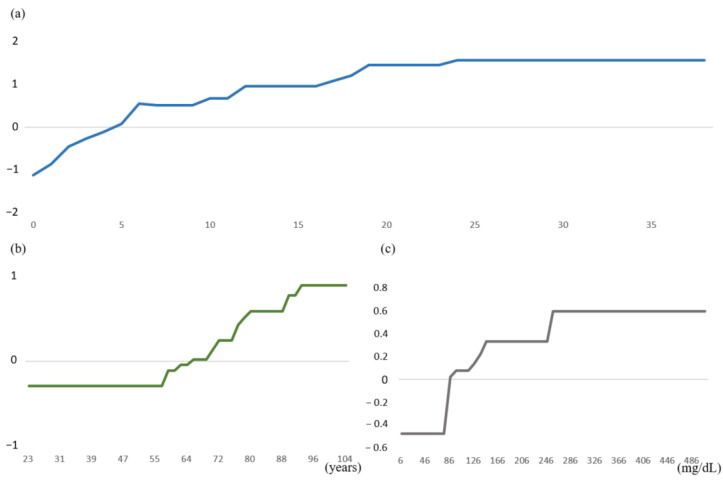
Partial dependence plot of variables. The *x*-axis represents the value of a particular variable; values on the *y*-axis indicate the risk of developing an unfavorable outcome. A positive value indicates that a higher variable value increases the risk of an unfavorable outcome. (**a**) NIHSS. (**b**) Age. (**c**) Fasting glucose.

**Figure 5 diagnostics-13-00842-f005:**
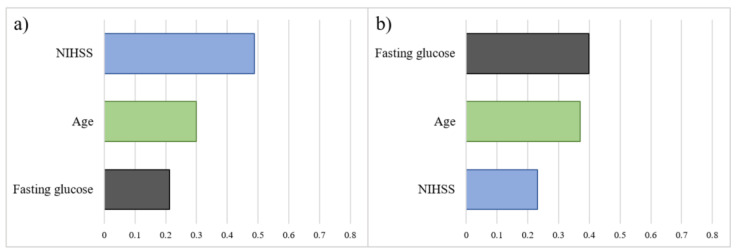
The importance and the ranking of the predictors of EVT with IVT and EVT without IVT subgroups. EVT, endovascular therapy; IVT, intravenous thrombolysis. (**a**) EVT with IVT. (**b**) EVT without IVT.

**Table 1 diagnostics-13-00842-t001:** Characteristics of patients with favorable and unfavorable 3-month outcomes after AIS.

	Favorable Outcome	Unfavorable Outcome	*p*-Value
*n* (%)	880 (47.62)	968 (52.38)	
Age (years)	63.73 (±12.43)	72.44 (± 13.41)	<0.0001 ***
Male, *n* (%)	609 (32.95)	533 (28.84)	<0.0001 ***
NIHSS score	3.27 (±3.4)	11.50 (± 8.66)	<0.0001 ***
Glycemic metrics			
Fasting glucose (mg/dL)	123.69 (±43.68)	140.70 (± 54.80)	<0.0001 ***
Glucose at admission (mg/dL)	158.30 (±71.73)	163.66 (± 80.67)	0.1746
HbA1c (%)	6.73 (±1.91)	6.69 (± 1.86)	0.6767
Vascular risk factors, *n* (%)			
Hypertension	607 (32.85)	699 (37.82)	0.1273
Diabetes mellitus	308 (16.67)	401 (21.70)	0.0046 **
Hyperlipidemia	621 (33.60)	603 (32.63)	0.0002 ***
Atrial fibrillation	102 (5.52)	236 (12.77)	<0.0001 ***
Previous stroke or TIA	83 (4.49)	188 (10.17)	<0.0001 ***
Ischemic heart disease	80 (4.33)	117 (6.33)	0.0371 *

*** *p* < 0.001; ** *p* < 0.01; * *p* < 0.05. Continuous data are presented as mean and standard deviation. Abbreviation: HbA1c, glycated hemoglobin; NIHSS, National Institute of Health Stroke Scale; TIA, transient ischemic attack.

## Data Availability

The data that support the findings of this study are available from Taiwan Stroke Registry (http://taiwanstrokeregistry.org/TSR/; accessed on 1 January 2022), but restrictions apply to the availability of these data, which were used under license for the current research and so are not publicly available.

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
