# Peer review of "XGBoost-Based Simple Three-Item Model Accurately Predicts Outcomes of Acute Ischemic Stroke"

_diagnostics, 2023, doi:10.3390/diagnostics13050842_

Round 1

Reviewer 1 Report

The paper sounds very interesting and has the potential to publish. The authors should revise the paper according to the following major comments: 

- The abstract section should present the main question of the paper. The method to solve this question and the main results, i.e., the abstract should contain the highlight of the paper. As the abstract presented in this form, it is not very clear what is the main question of the paper. 

- In the introduction section the authors repeated the references [1, 2] 3 times in 4 lines. The references can be cited at the end of the first section. 

- Nomenclature must be added to the paper. As well as all the physical/mathematical units of each variable and parameter. 

-The connection between sections 2 and 3 is not clear. How did the authors apply the mathematical expression in section 2 to obtanied the results present in section 3? This connection is critical to understand the results of the paper.

- The following papers can be added: 

1: Su, P.-Y., Wei, Y.-C., Luo, H., Liu, C.-H., Huang, W.-Y., Chen, K.-F., Lin, C.-P., Wei, H.-Y., & Lee, T.-H. (2022). Machine Learning Models for Predicting Influential Factors of Early Outcomes in Acute Ischemic Stroke: Registry-Based Study. In JMIR Medical Informatics (Vol. 10, Issue 3, p. e32508). JMIR Publications Inc. https://doi.org/10.2196/32508

2: Nave, Op., & Elbaz, M. (2021). Artificial immune system features added to breast cancer clinical data for machine learning (ML) applications. In Biosystems (Vol. 202, p. 104341). Elsevier BV. https://doi.org/10.1016/j.biosystems.2020.104341

- Figures 1, and 3 should be explained, please. 

- The mathematical expressions presented in section 2 are not clear. Please explain every equation and more critically explain the connections between the present expressions.   

- The conclusion section must be extended. 

Author Response

February 19, 2023

Dear Professor Lung Chan,

Editor of Diagnostics:

Re: Submission of Revised Manuscript

Thank you for the opportunity to submit a revised version of our manuscript titled “XGBoost-based Simple Three-item Model Accurately Predicts Outcomes of Acute Ischemic Stroke” to Diagnostics.

The authors would like to thank the editor and the reviewers for the thoughtful

evaluation of our manuscript. We have carefully revised the paper following your valuable comments and suggestions, and we believe that the new version is now

greatly improved concerning its content and relevance to your readers.

We comprehensively address each of the reviewers’ concerns as follows. Thank you for considering our manuscript for publication in Diagnostics.

Best regards,

Point-by-point response to Reviewer 1’s comments:

The paper sounds very interesting and has the potential to publish. The authors should revise the paper according to the following major comments: 

Comment 1: The abstract section should present the main question of the paper. The method to solve this question and the main results, i.e., the abstract should contain the highlight of the paper. As the abstract presented in this form, it is not very clear what is the main question of the paper. 

Response 1: We thank the reviewer for the insightful comments. In our revised manuscript, based on the reviewer’s suggestions, we have revised our abstract to highlight the main findings and importance of the paper.

Please see our revised Abstract on Page 1, Lines 21-35:

Abstract: An all-inclusive and accurate prediction of outcomes for patients with acute ischemic stroke (AIS) is crucial for clinical decision-making. This study developed extreme gradient boosting (XGBoost)-based models using three simple factors: age, fasting glucose, and National Institutes of Health Stroke Scale (NIHSS) scores to predict the three-month functional outcomes after AIS. We retrieved the medical records of 1,848 patients diagnosed with AIS and managed at a single medical center between 2016 and 2020. We developed and validated the predictions and ranked the importance of each variable. The XGBoost model achieved notable performance, with an area under the curve of 0.8595. As predicted by the model, the patients with initial NIHSS score>5, aged over 64 years, and fasting blood glucose>86 mg/dL were associated with unfavorable prognoses. For patients receiving endovascular therapy, fasting glucose was the most important predictor. The NIHSS score at admission was the most significant predictor for those who received other treatments. Our proposed XGBoost model shows a reliable predictive power of AIS outcomes using readily available and simple predictors, and also demonstrates the validity of the model for application in patients receiving different AIS treatments, providing clinical evidence for future optimization of AIS treatment strategies.”

Comment 2: In the introduction section the authors repeated the references [1, 2] 3 times in 4 lines. The references can be cited at the end of the first section. 

Response 2: We thank the reviewer for the suggestions. We have modified the introduction to avoid duplication of references.

Please kindly see our revised first section of the introduction on Pages 1-2, Lines 39-53:

“Stroke is the second leading cause of death and the third leading cause of disability worldwide [1, 2]. The global burden of stroke has increased substantially over the past 30 years owing to population growth, aging, and exposure to risk factors such as high plasma levels of fasting glucose. Among all stroke types, ischemic strokes account for the largest proportion of all new strokes. In 2019, there were an estimated 7.6 million new ischemic strokes globally, and the economic burden of stroke is estimated to be more than US$ 721 billion [1]. In addition to the high incidence of stroke occurrence, recurrence, and death, ischemic stroke may cause various functional impairments, cognitive impairment, or post-stroke depression, contributing to the high prevalence of disability in stroke survivors [1-5]. In the report by Yao et al. [4], two-thirds of patients with ischemic stroke had sequelae symptoms, and nearly one-fourth had varying degrees of disability. Consequently, stroke has a wide range of negative physical and economic impacts on patients and their families. Metabolic factors such as high blood pressure, high blood sugar, and high cholesterol are strongly linked to ischemic stroke. These potentially modifiable risks are associated with outcomes after the episode of ischemic stroke [1-3].”

Comment 3: Nomenclature must be added to the paper. As well as all the physical/mathematical units of each variable and parameter. 

Response 3: Thank you for your reminder. We have added the unit to each variable accordingly, as suggested.

Comment 4: The connection between sections 2 and 3 is not clear. How did the authors apply the mathematical expression in section 2 to obtanied the results present in section 3? This connection is critical to understand the results of the paper.

Response 4: Thank you for your kind reminder. We have added the paragraph on Page 5, Lines 189-196:

“In the section of Result, we used the packages in the R environment. Ten-fold cross-validation is a method in the package “caret”, which can be used to find the optimal hyperparameters for the prediction model in terms of AUC. The optimal hyperparameters are decided during ten-fold cross-validation and incorporated while building the prediction model based on the XGBoost algorithm. We used the indicator “Gain” to rank the importance of the variables in different treatments. To find the relationship between 3-month unfavorable outcome and each variable, partial dependence plots are used to depict the propensity in all variables.”

Comment 5: The following papers can be added: 

1: Su, P.-Y., Wei, Y.-C., Luo, H., Liu, C.-H., Huang, W.-Y., Chen, K.-F., Lin, C.-P., Wei, H.-Y., & Lee, T.-H. (2022). Machine Learning Models for Predicting Influential Factors of Early Outcomes in Acute Ischemic Stroke: Registry-Based Study. In JMIR Medical Informatics (Vol. 10, Issue 3, p. e32508). JMIR Publications Inc. https://doi.org/10.2196/32508

2: Nave, Op., & Elbaz, M. (2021). Artificial immune system features added to breast cancer clinical data for machine learning (ML) applications. In Biosystems (Vol. 202, p. 104341). Elsevier BV. https://doi.org/10.1016/j.biosystems.2020.104341

Response 5: We thank the reviewer for this comment. We have added these two articles to our references. Please kindly see references 8 and 11:

8. Nave, O.; Elbaz, M. Artificial immune system features added to breast cancer clinical data for machine learning (ML) applications. Biosystems 2021, 202, 104341, doi:10.1016/j.biosystems.2020.104341.

11. Su, P.Y.; Wei, Y.C.; Luo, H.; Liu, C.H.; Huang, W.Y.; Chen, K.F.; Lin, C.P.; Wei, H.Y.; Lee, T.H. Machine Learning Models for Predicting Influential Factors of Early Outcomes in Acute Ischemic Stroke: Registry-Based Study. JMIR Med Inform 2022, 10, e32508, doi:10.2196/32508.

Comment 6: Figures 1, and 3 should be explained, please. 

Response 6: Thank you for your reminder. We have added the paragraph on Page 5, Lines 198-204:

“The analytical flowchart of this study is shown in Fig. 1. Data from 2,384 participants were collected. After the completion of screening data, the records of 1,848 patients were analyzed. There are several hyperparameters that should be controlled. To achieve the best predictive performance, hyperparameters need to be optimal. Ten-fold cross-validation is used to find the optimal hyperparameters to improve the prediction model. Finally, the ranking importance of variables and PDPs are generated by the final prediction model to uncover the relationship between 3-month un-favorable outcome and variables.”

We also revised the paragraph on Page 7, Lines 243-248:

“Fig. 3 illustrates the importance of predictors for the entire cohort and patients receiving different treatments. The importance is measured by “Gain,” which is generated from the package “xgboost” from the ranking based on “Gain.” In both the whole cohort and patients who received IVT or non-thrombolytic treatment, the NIHSS score was the most crucial variable for predicting AIS outcome. In the EVT group, the most important predictor was fasting glucose level, with the NIHSS score being less important than age.”

Comment 7: The mathematical expressions presented in section 2 are not clear. Please explain every equation and more critically explain the connections between the present expressions.   

Response 7: Thank you for your comment. We have revised the section accordingly on Page 4, Lines145-166.

“All XGBoost models were developed using the “caret” and “xgboost” packages in RStudio software (version 1.3.1073; 2009 – 2020 RStudio, PBC). XGBoost algorithm was used to train and predict 3-month unfavorable outcome.

 (Please refer to our attached pdf file for the formula.)          (1)

In the process of training prediction model, K trees are generated. In XGBoost, the set of functions used in the model was minimized by the following regularized objective:

(Please refer to our attached pdf file for the formula.)                        (2)

where , l is a differentiable convex loss function measuring the predicted outcome , and the real outcome yi. Considering that the tree with more depth smoothens the final learned weights, the prediction model is penalized by the regularization term , used to avoid overfitting.

XGBoost will not generate all trees at one time. It starts from the 0-th tree. It split each node in the tree, the original node continues splitting to right and left node. After a new node splits, it is essential to check whether the new split will give ‘Gain’ to the loss function. The variable and its point with maximum value of ‘Gain’ will be selected.

The final information ‘Gain’ of the objective function after each split is:

(Please refer to our attached pdf file for the formula.)         (3)

where, L and R are the subsets of left and right nodes after splitting from the instance set I.  is a splitting threshold for suppressing the growth of the tree. It is an important metric to rank the importance of variables in this study.”

Comment 8: The conclusion section must be extended.

Response 8: We thank the reviewer for the suggestion. We have revised our conclusions to strengthen the description of the significance of the findings and results of this study. Please kindly refer to Page 11, Lines 372-386:

“In the present study, we developed an accurate and reliable model to predict outcomes at three months following AIS by applying XGBoost-based machine learning algorithms using three predictors: age, fasting glucose, and NIHSS score. The generated XGBoost model achieved a notable validation performance with an AUC of 0.8595. This model also demonstrates its validity in patients with AIS who received different treatments, including IVT, EVT, and non-thrombolytic therapy. In the whole cohort and patients who received IVT or non-thrombolytic treatment, the NIHSS score was the most crucial predictor of AIS outcome. In the EVT group, the most important predictor was fasting glucose. This model also demonstrates its validity in patients with AIS who received different treatments, including IVT, EVT, and non-thrombolytic therapy. Therefore, our results show the potential clinical applicability of this model to facilitate individualized risk prediction and aid the formulation of diagnostic decisions for patients with AIS. Based on these results, future work is being shaped to develop predictive models that incorporate a more diverse patient population and can automatically update with new patient information and future advancements.”

We thank the reviewers again for all the valuable comments and suggestions.

Reviewer 2 Report

Authors aim to predict the outcomes of stroke by employing a machine learning algorithm – XGBOOS based on three parameters – age, NIHSS and glucose. Accordingly, NIHSS was found to be the most important predictor of unfavorable outcome. The manuscript lacks a clear hypothesis and requires more methodological details to be provided for clarity.

Major points

1. Introduction. Please specify the hypothesis and the aims of this study.

2. Materials and Methods. 2.1. Participants. Authors do not mention anything about patients‘ comorbidities as an exclusion/inclusion criterion?  

3. Materials and Methods. 2.2. Statistical analyses. What does ‚All independent variables‘ mean? What are these variables and within each statistical model they are indepedent?

4. Materials and Methods. 2.4. Ten-fold cross-validation. How the names of hyperparameters are derived? For example, what ‚min_child_weight‘ or ‚gamma‘ fort he current study aims?

5. Results. 3.2. Model performance. Line 185. Accuracy is indicated here 0.7862 but in Table S1 0.7654. which one is correct?

6. Results. Figure 3. Do ‚gain‘ has an unit?

7. Results. Figure 3. The values of x-axis should be all the same for a), b), c), and d); that is 0.8.

8. Results. Line 212. ‚to have adverse outcomes after AIS‘. What is adverse outcome? This term was not used before.

9. Results. Authors wrote in the statistical section that ‚Fisher’s exact test was used to determine associations between the two categorical variables in patients with favorable and unfavorable outcomes‘. However, in the Results, nothin is mentioned on any associations.

10. Results. What is the relevance of the indicated values of hyperparameters fort he obtained results from the prediction models. What do these values mean?

11. Results. Since the group of EVT contains and patients with IVT, it is difficult to interpret the accuracy of the model for this group. Is there any chance to analyse patients, which did not have IVT prior to EVT?

12. Informed Consent Statement is missing

13. Data Availability Statement is missing

14. Author Contributions is missing

15. Abstract is missing

Minor points

1. Introduction. Line 61. Please define ADL  

2. Results. 3.1. Demographical and baseline characteristics of the AIS cohort. Line 163. NIHSS of 7.5 [± 7.9] years?

3. Results. Figure 1. Please revise this sentence ‚The participants who didnt be recorded fully‘ for grammar.

4. Results. Figure 1. The direction of the arrow should be opposite fort he exclusion box

5. Discussion. Line 227. In Results it was mentioned NIHSS of 5 but here 10.

Author Response

February 19, 2023

Dear Professor Lung Chan,

Editor of Diagnostics:

Re: Submission of Revised Manuscript

Thank you for the opportunity to submit a revised version of our manuscript titled “XGBoost-based Simple Three-item Model Accurately Predicts Outcomes of Acute Ischemic Stroke” to Diagnostics.

The authors would like to thank the editor and the reviewers for the thoughtful

evaluation of our manuscript. We have carefully revised the paper following your valuable comments and suggestions, and we believe that the new version is now

greatly improved concerning its content and relevance to your readers.

We comprehensively address each of the reviewers’ concerns as follows. Thank you for considering our manuscript for publication in Diagnostics.

Best regards,

Point-by-point response to Reviewer 2’s comments:

Comments and Suggestions for Authors

Authors aim to predict the outcomes of stroke by employing a machine learning algorithm – XGBOOS based on three parameters – age, NIHSS and glucose. Accordingly, NIHSS was found to be the most important predictor of unfavorable outcome. The manuscript lacks a clear hypothesis and requires more methodological details to be provided for clarity.

Major points

Comment  1:Introduction. Please specify the hypothesis and the aims of this study.

Response 1: We thank the reviewer for the comment. We have modified our introduction to specify this study’s hypothesis and aims. Please kindly see Page 2, Lines 93-97:

“The current study hypothesized that age, fasting glucose, and NIHSS score as simple, generic, and readily available factors that provide sufficient predictive power to predict outcomes after stroke and aimed to develop predictive models using these most general predictors, borrowing the advantages of machine learning algorithms in analyzing large and nonlinear data.”

Comment 2: Materials and Methods. 2.1. Participants. Authors do not mention anything about patients‘ comorbidities as an exclusion/inclusion criterion?  

Response 2: We thank the reviewer for the suggestion. In our revised manuscript, we added the description of patients‘ comorbidities in the Materials and Methods section. Please kindly see Page 3, Lines 120-125:

“Demographic data were collected at admission, including age, sex, NIHSS score, and treatment for AIS (i.e., IVT, EVT, and non-thrombolysis). IVT and EVT treatments followed the American Heart Association/American Stroke Association guidelines. The patients’ vascular risks and comorbidities, including hypertension, diabetes mellitus, hyperlipidemia, atrial fibrillation, previous stroke or transient ischemic attack (TIA), and ischemic heart disease were also documented.”

Comment 3: Materials and Methods. 2.2. Statistical analyses. What does ‚All independent variables‘ mean? What are these variables and within each statistical model they are indepedent?

Response 3: Independent variables are the predictors influencing the outcome in the prediction model. We have revised the sentence to improve readability on Page 3, Lines 136-137:

“All continuous variables were expressed as mean and standard deviation.”

Comment 4: Materials and Methods. 2.4. Ten-fold cross-validation. How the names of hyperparameters are derived? For example, what ‚min_child_weight‘ or ‚gamma‘ for the current study aims?

Response 4: Thank you for your question. These hyperparameters were optimized during the process of ten-fold cross-validation automatically. The hyperparameters which are tuned commonly in the XGBoost algorithm define below:

  • nround(range: 0-inf): the number of trees generated.
  • eta(range:0-1): Learning rate is the rate the model learns the patterns from data.
  • gamma(range: 0-inf): control regulations to avoid overfitting.
  • max_depth(range: 0-inf): the depth of trees.
  • mid_child_weight(range: 0-inf): a tree stops splitting while the sum of instance weight of leaf lower node than mid_child_weight.
  • subsample(range:0-1): the number of observations sampled to a tree.
  • colsample_bytree(range:0-1): the number of variables sampled to a tree.

Hyperparameter tuning aims to find the optimal hyperparameters for predicting the 3-month outcome, which can help the prediction model achieve the best ROC and lowest error.

Comment 5: Results. 3.2. Model performance. Line 185. Accuracy is indicated here 0.7862 but in Table S1 0.7654. which one is correct?

Response 5: Thank you for your reminder. The accuracy in Table S1 is correct. We have modified in Manuscript for Revision.

Comment 6: Results. Figure 3. Do ‚gain‘ has an unit?

Response 6: Thank you for your question. ‘Gain’ doesn’t have a unit. It is a metric evaluating the relative contribution of the corresponding variable to the model.

Comment 7: Results. Figure 3. The values of x-axis should be all the same for a), b), c), and d); that is 0.8.

Response 7: Thank you for your valuable comment. We have revised Figure 3 accordingly, as suggested.

Comment 8: Results. Line 212. ‚to have adverse outcomes after AIS‘. What is adverse outcome? This term was not used before.

Response 8: We appreciate the careful review of the reviewers. We have corrected the “adverse outcome” to “unfavorable outcome” to maintain consistency in the wording of the article.

Comment 9: Results. Authors wrote in the statistical section that ‚Fisher’s exact test was used to determine associations between the two categorical variables in patients with favorable and unfavorable outcomes‘. However, in the Results, nothin is mentioned on any associations.

Response 9: Thank you for your valuable comments. We have added the statement on Page 5, Lines 214-217:

“Furthermore, in vascular risk factors, Fisher’s exact test also shows that diabetes mellitus, hyperlipidemia, atrial fibrillation, previous stroke or TIA, and ischemic heart disease is associated with 3-month outcome, and the gender of patients has association to 3-month outcome as well.”

Comment 10: Results. What is the relevance of the indicated values of hyperparameters for the obtained results from the prediction models. What do these values mean?

Response 10: Thank you for your question. These values of hyperparameters are generated by ten-fold cross-validation. The prediction model with these optimal hyperparameter values can achieve the best performance. The definitions of hyperparameters are described below:

  • nround(range: 0-inf): the number of trees generated.
  • eta(range:0-1): Learning rate is the rate the model learns the patterns from data.
  • gamma(range: 0-inf): control regulations to avoid overfitting.
  • max_depth(range: 0-inf): the depth of trees.
  • mid_child_weight(range: 0-inf): a tree stops splitting while the sum of instance weight of leaf lower node than mid_child_weight.
  • subsample(range:0-1): the number of observations sampled to a tree.
  • colsample_bytree(range:0-1): the number of variables sampled to a tree.

Comment 11: Results. Since the group of EVT contains and patients with IVT, it is difficult to interpret the accuracy of the model for this group. Is there any chance to analyse patients, which did not have IVT prior to EVT?

Response 11: We concur with the reviewer’s opinion. In the revised manuscript, to further interpret the EVT group, we compared the characteristics and assessed the importance of predictors of the two subgroups separately. Please kindly see Page 9, Lines 266-278, and Figure 5:

“3.3. Comparison of the patients with and without IVT in the EVT group

To further interpret the current model for the EVT group, we compared the characteristics of patients in the EVT group with and without IVT and assessed the importance of predictors of the two subgroups separately. When comparing the EVT with and without IVT subgroups, there was no difference between the age (67.12 [± 12.48] vs. 70.28 [± 13.75], p=0.2535), fasting glucose (184.15 [± 56.61] vs. 168.60 [± 58.16], p=0.2049) or NIHSS scores (18.15 [± 6.60] vs. 19.04 [± 7.60], p=0.5473). There was also no difference in the outcomes by mRS assessment between the two subgroups three months after the stroke (p=1.000).

Fig. 5 demonstrates the importance of predictors for the EVT group with and without IVT treatment. Similar to the non-thrombolytic and IVT groups, the NIHSS score was the most important predictor of prognostic relevance in the EVT with IVT subgroup (Fig. 5a). However, in the subgroup of EVT without IVT, fasting glucose was determined as the most crucial factor in predicting outcomes in the XGBoost model (Fig. 5b).”

Comment 12: Informed Consent Statement is missing

Response 12: Thank you for your reminder. We have added Informed Consent Statement accordingly. Please see our Informed Consent Statement on Page 11-12, Lines 405-408:

Informed Consent Statement: The study involving human participants were reviewed and approved by Joint Institutional Review Board of Taipei Medical University. Written informed consent for participation was not required for this study in accordance with the national legislation and the institutional requirements.”

Comment 13: Data Availability Statement is missing

Response 13: Thank you for your reminder. We have added Data Availability Statement accordingly. Please see our Data Availability Statement on Page 12, Lines 409-412:

Data Availability Statement: The data that support the findings of this study are available from Taiwan Stroke Registry (http://taiwanstrokeregistry.org/TSR/), but restrictions apply to the availability of these data, which were used under license for the current research and so are not publicly available.“

Comment 14: Author Contributions is missing

Response 14: Thank you for your reminder. We have added Author Contributions accordingly. Please see our Author Contributions on Page 11, Lines 393-398:

Author Contributions: Conceptualization, C.-C.C. and C.-Y.K.; methodology, E.C.-Y.S. and C.-Y.K.; software, C.-Y.K.; validation, E.C.-Y.S.; formal analysis, C.-Y.K.; investigation, C.-C.C. and J.-H.C.; resources, C.-C.C. and J.-H.C.; data curation, C.-C.C. and J.-H.C.; writing—original draft preparation, C.-C.C. and C.-Y.K.; writing—review and editing, C.-C.C., E.C.-Y.S. and C.-Y.K.; visualization, Y.-T.C. and C.-Y.K.; supervision, C.-C.C. and C.-Y.K.; project administration, C.-C.C.; funding acquisition, C.-C.C. All authors have read and agreed to the published version of the manuscript.”

Comment 15: Abstract is missing

Response 15: Thank you for your reminder. We have added Abstract accordingly. Please see our revised Abstract on Page 1, Lines 21-35:

Abstract: An all-inclusive and accurate prediction of outcomes for patients with acute ischemic stroke (AIS) is crucial for clinical decision-making. This study developed extreme gradient boosting (XGBoost)-based models using three simple factors: age, fasting glucose, and National Institutes of Health Stroke Scale (NIHSS) scores to predict the three-month functional outcomes after AIS. We retrieved the medical records of 1,848 patients diagnosed with AIS and managed at a single medical center between 2016 and 2020. We developed and validated the predictions and ranked the importance of each variable. The XGBoost model achieved notable performance, with an area under the curve of 0.8595. As predicted by the model, the patients with initial NIHSS score>5, aged over 64 years, and fasting blood glucose>86 mg/dL were associated with unfavorable prognoses. For patients receiving endovascular therapy, fasting glucose was the most important predictor. The NIHSS score at admission was the most significant predictor for those who received other treatments. Our proposed XGBoost model shows a reliable predictive power of AIS outcomes using readily available and simple predictors, and also demonstrates the validity of the model for application in patients receiving different AIS treatments, providing clinical evidence for future optimization of AIS treatment strategies.”

Minor points

Comment 1:  Introduction. Line 61. Please define ADL  

Response 1: We are thankful to the reviewers for the careful review. We have corrected the “ADL” to “activities of daily living.” Please see Page 2, Line 78.

Comment 2:  Results. 3.1. Demographical and baseline characteristics of the AIS cohort. Line 163. NIHSS of 7.5 [± 7.9] years?

Response 2: Thank you for your valuable comment. We have removed “years” in the sentence.

Comment 3:  Results. Figure 1. Please revise this sentence ‚The participants who didnt be recorded fully‘ for grammar.

Response 3: Thank you for your comment. We have modified the sentence to “The participants with missing information in any of the following variables:” in Figure 1.

Comment 4:  Results. Figure 1. The direction of the arrow should be opposite for the exclusion box.

Response 4: Thank you for your valuable comment. We have revised Figure 1 accordingly as suggested.

Comment 5:  Discussion. Line 227. In Results it was mentioned NIHSS of 5 but here 10.

Response 5: Thank you for your reminder. We have revised in the manuscript.

We thank the reviewers again for all the valuable comments and suggestions.

Round 2

Reviewer 1 Report

I have read the paper again and the authors revise the paper according to my comments. 

Please before publication, send the paper to English editor

Good luck 

Reviewer 2 Report

The authors replied to raised concerns.